# Molecular Cloning and Functional Characterization of Galectin-1 in Yellow Drum (*Nibea albiflora*)

**DOI:** 10.3390/ijms24043298

**Published:** 2023-02-07

**Authors:** Baolan Wu, Qiaoying Li, Wanbo Li, Shuai Luo, Fang Han, Zhiyong Wang

**Affiliations:** Key Laboratory of Healthy Mariculture for the East China Sea, Ministry of Agriculture and Rural Affairs, Fisheries College, Jimei University, Xiamen 361021, China

**Keywords:** galectin-1, agglutination, antibacterial activity, innate immunity, *Nibea albiflora*

## Abstract

Galectins are proteins that are involved in the innate immune response against pathogenic microorganisms. In the present study, the gene expression pattern of *galectin-1* (named as *NaGal-1*) and its function in mediating the defense response to bacterial attack were investigated. The tertiary structure of *Na*Gal-1 protein consists of homodimers and each subunit has one carbohydrate recognition domain. Quantitative RT-PCR analysis indicated that *NaGal-1* was ubiquitously distributed in all the detected tissues and highly expressed in the swim-bladder of *Nibea albiflora*, and its expression could be upregulated by the pathogenic *Vibrio harveyi* attack in the brain. Expression of *Na*Gal-1 protein in HEK 293T cells was distributed in the cytoplasm as well as in the nucleus. The recombinant *Na*Gal-1 protein by prokaryotic expression could agglutinate red blood cells from rabbit, *Larimichthys crocea*, and *N. albiflora*. The agglutination of *N. albiflora* red blood cells by the recombinant *Na*Gal-1 protein was inhibited by peptidoglycan, lactose, D-galactose, and lipopolysaccharide in certain concentrations. In addition, the recombinant *Na*Gal-1 protein agglutinated and killed some gram-negative bacteria including *Edwardsiella tarda*, *Escherichia coli*, *Photobacterium phosphoreum*, *Aeromonas hydrophila*, *Pseudomonas aeruginosa*, and *Aeromonas veronii*. These results set the stage for further studies of *Na*Gal-1 protein in the innate immunity of *N. albiflora*.

## 1. Introduction

Lectins are pattern recognition receptors (PRRs) with carbohydrate recognition domains (CRDs) and widely found in vertebrate tissues [1]. They specifically bind to pathogen-associated molecular patterns (PAMPs) and play important roles in host defense against various pathogenic microbial invasions and infections [2,3]. Galectins belong to one of the major lectin families and share several common characteristics [4,5]: metal-independent activity, high affinity for *β*-galactosides, core subdivided into three types: the prototype containing one CRD and tending to form homodimers, the chimera type consisting of an N-terminus rich in proline and glycine and a C-terminus CRD, and the tandem-repeat type containing two corresponding CRDs. To date, galectins have been detected and isolated from various fish, and these fish galectins exist in various cells and have been identified to mediate diverse biological processes involved in the regulation of innate immune responses [2,6], such as activation of cellular interactions, pathogen recognition, bacterial agglutination and killing, antiviral process, and modulation of immune responses.

Galectin-1 is a prototype member of galectin family and was also the first galectin to be discovered [7]. Galectin-1 is an important immunoregulatory factor [8], which has been proved to regulate the physiological function of monocyte and macrophage [9]. In addition, Cedeno-Laurent, F. et al. showed that galectin-1 could bind to T cell membrane glycoproteins and potentiate the immunoregulatory function of T cells [10]. So far, in the classification of galectins in mammals, the identified prototypes include galectin-1, -2, -5, -7, -10, -11, -13, -14, and -15; the reported fish prototype galectins are mainly galectin-1 [11] and -2 [12], which, like other galectins from fish (such as: galectin-3, -8, -9), can participate in the immune response of fish. For example, the galectin-2 protein of *Oreochromis niloticus* enhanced the phagocytosis of macrophages [13]. The galectin-1 protein from *Oplegnathus fasciatus* had significant antiviral activity against rock bream irido virus [14]. The *Solen grandis* galectin-1 protein not only promoted the phagocytosis of hemocytes against *E. coli* in vitro but also directly participated in the process of immune regulation [15]. The recombinant galectin-1 protein of *Paralichthys olivaceus* could neutralize the lymphocystic disease virus and inhibit the formation of cytopathic effects, and had potential anti-inflammatory activity [16]. Poisa-Beiro, L et al. screened a *galectin-1* gene related to immune defense from *Dicentarchus labrax* against nodavirus, indicating that it had a potential anti-inflammatory activity [17].

The yellow drum *(Nibea albiflora)*, an important fish species of Sciaenidae family, is of great economic value in China [18]. In recent years, with the rapid development of marine aquaculture, *Vibrios* diseases have seriously affected the *N. albiflora* aquaculture industry, and cause heavy economic losses [19]. However, little is known about the pathogenic mechanisms in *N. albiflora*, therefore, the studies of immune-related genes are imperative. In the present study, we analyzed the molecular structure of a galectin-1 from *N. albiflora* (named as *NaGal-1*) and studied its possible immune function, to provide reference for further research on the important role of galectins in innate immunity of *N. albiflora*.

## 2. Results

### 2.1. Sequence Characteristic Analysis of NaGal-1

The genomic DNA sequence of *NaGal-1* was 4412 bp in length, containing 4 exons and 3 introns (Figure 1A). The 3 introns were 2400, 712, and 337 bp in length, respectively. All introns conformed to the typical intron-splicing motif (GT/AG rule). The cDNA sequence of *NaGal-1* was 963 bp in length, including 78 bp 5’ UTR, 477 bp 3’ UTR, and contained an open reading frame of 408 bp encoding a protein of 135 amino acid residues. The predicted molecular mass of the *Na*Gal-1 polypeptide was 15.26 kDa, and the isoelectric point was 5.59. NetPhos 3.1 Server predicted that there were 10 phosphorylation sites in *Na*Gal-1: 5 serine (Ser), 3 threonine (Thr), and 2 tyrosine (Tyr) (Figure 1B). Neither signal peptide nor transmembrane region was predicted by SignaIP-5.0 Server and TMHMM Server v.2.0, respectively. *Na*Gal-1 protein had a carbohydrate recognition domain (CRD), tertiary structure of *Na*Gal-1 protein is a homodimers formed through hydrophobic interactions between N- and C- terminus residues of two subunits related by a 2-fold rotation axis perpendicular to the plane of the two *β*-sheets (Figure 1C) [20].

### 2.2. Multiple Sequence Alignments and Phylogenetic Analysis of NaGal-1

The multiple sequence alignment and phylogenetic analysis of *Na*Gal-1 protein (GemBank No: OQ139528) were performed based on the Clustal Omega. The results revealed significant protein sequence similarity between *Na*Gal-1 and Gal-1 of other vertebrates, and *Na*Gal-1 displayed the highly similarity to the Gal-1 proteins of *L. crocea* (90.37%) and *Collichthys lucidus* (85.93%) (Table 1). There were two highly conserved sugar binding motifs H-NPR and W--E-R (Figure 2). The evaluated molecular evolutionary relationship of *Na*Gal-1 was analyzed by phylogenetic tree, and the results showed that Gal-1 of fishes, amphibians, birds, and mammals formed four clusters (the accession number of each species in Table 1), and *Na*Gal-1 protein was closely related to Gal-1 of *L. crocea* and *C. lucidus* (Figure 3), which was generally in agreement with the traditional taxonomy.

### 2.3. Tissue Distribution of NaGal-1 and Response to V. harveyi Infection

The distribution of *NaGal-1* mRNA transcripts in 12 different tissues and the mRNA expression of *NaGal-1* after infection with *V. harveyi* in four kinds of tissues were detected using Quantitative real-time PCR (qRT-PCR), and *β-actin* gene as an internal control. As shown in Figure 4A, *NaGal-1* is ubiquitously distributed in various tissues, but there are some differences in expression level. Among 12 tissues, the highest expression was in the swim-bladder and gill. After infection with *V. harveyi*, the expressions of *NaGal-1* in brain and the classical immune tissues head kidney, liver, and spleen are shown in Figure 4B. In the brain, the expression of *NaGal-1* increased significantly during 6–96 h. Compared to the control, there was no significant change in the expression of *NaGal-1* in the head kidney and spleen. In the liver, the expression of *NaGal-1* was down-regulated during 6–96 h after stimulation with *V. harveyi*.

### 2.4. Expression of NaGal-1 in HEK 293T Cells

In order to determine the expression of *Na*Gal-1 protein in HEK 293T Cells, the eukaryotic expression vector *NaGal-1-EGFP* was constructed and transfected into HEK 293T cells. At the same time, the *pEGFP-N1* was transfected as a negative control. The result of Western blot showed the successful expression of *Na*Gal-1-GFP protein in HEK 293T cells, which was consistent with its actual size (41.26 kDa, Figure 5A). Expression of *Na*Gal-1-GFP protein was found in both cytoplasm and nucleus, which was similar to that of EGFP protein (Figure 5B).

### 2.5. Prokaryotic Expression and Purification of NaGal-1

In order to study the function of *Na*Gal-1, the *NaGal-1* was ligated into vector *pET-32a* for prokaryotic expression. As shown in Figure 6, Trx-*Na*Gal-1 protein and Trx tag protein were expressed in *E. coli* BL21 (DE3) and purified by chromatography on Ni SepharoseTM 6 Fast Flow (GE Healthcare). The obtained proteins were analyzed by SDS-PAGE. Coomassie brilliant blue staining showed that most of soluble proteins were present in the supernatant, and the recombinant Trx-*Na*Gal-1 protein was 33.26 kDa and the Trx tag was 18 kDa, corresponding to the predicted size exactly.

### 2.6. Hemagglutination and Sugar Inhibition Assay of NaGal-1

We performed a hemagglutination test of the purified recombinant protein, and the results showed that the purified Trx-*Na*Gal-1 protein could agglutinate red blood cells (RBCs) from rabbit, *L. crocea*, and *N. albiflora*, and the RBC agglutination efficiency was not influenced by the addition of Ca^2+^. No RBC agglutination was observed in Trx protein control groups under the same conditions (Figure 7A).

The inhibitory effect of sugars on hemagglutination was detected to identify the carbohydrate binding specificity of *Na*Gal-1 protein. The results revealed that agglutinating activity of Trx-*Na*Gal-1 protein toward *N. albiflora* RBCs was inhibited by lipopolysaccharide (LPS), peptidoglycan (PGN), lactose, and D-galactose. Other sugars including D-mannose, D-(+)-raffinose pentahydrate, D-(+)-maltose monohydrate, D-fructose, D-(+)-trehalose dihydrate, D-glucose, and D-(+)-sucrose showed no inhibitory effect on hemagglutination. In particular, the minimum inhibitory concentrations for LPS, PGN, lactose, and D-galactose were 0.38 ng/mL, 1.52 ng/mL, 2.44 µg/mL, and 19.53 µg/mL, respectively. The results indicated that *Na*Gal-1 protein could specifically bind to LPS, PGN, lactose, and D-galactose (Figure 7B).

### 2.7. Bacterial Agglutination and Antibacterial Activity of NaGal-1

Live/Dead^®®^BacLight™Bacterial Viability Kit was used to study the bacterial agglutination and antibacterial activity of Trx-*Na*Gal-1 protein. The SYTO9 stain labels bacteria with both intact and damaged cell wall and the bacteria show green fluorescence, while the propidium iodide (PI) stain only labels bacteria with damaged cell wall and the dead bacteria show red fluorescence. In Figure 8, after treatment with Trx-*Na*Gal-1, the six species of bacteria were agglutinated, including *Edwardsiella tarda*, *Escherichia coli*, *Photobacterium phosphoreum*, *Aeromonas hydrophila*, *Pseudomonas aeruginosa*, and *Aeromonas veronii*. They show green and red fluorescence, indicating that the bacteria cell wall was disrupted after being treated with Trx-*Na*Gal-1 and the bacteria were killed. While, there was no obvious antibacterial activity of Trx-*Na*Gal-1 against other six species of bacteria including *Pseudomonas plecoglossicida*, *Vibrio parahemolyticus*, *V. harveyi*, *Pseudomonas putida*, *V. alginolyticus*, and *V. vulnificus*. In addition, bacteria treated with Trx protein showed only green fluorescence, indicating that Trx protein has no antibacterial effect on bacteria. Taken together, NaGal-1 protein could agglutinate and kill bacteria in a targeted way with narrow spectra.

## 3. Discussion

In this study, we identified a *galectin-1* (named *NaGal-1*) from *N. albiflora* and analyzed its sequence by bioinformatics. The results showed that *Na*Gal-1 had the main characteristics of galectin: no signal peptide, no transmembrane region, including a kind of CRD, with highly conserved sugar binding motifs (H-NPR and W--E-R). The highly conserved sugar binding motif is the key to the agglutination activity of galectins. It was reported that homodimers were formed through the interaction between hydrophobic surfaces presented by each subunit when galectin-1 performed its function. *Na*Gal-1 had highly similarity with galectin-1 of *L. crocea* and *C. lucidus* (Table 1). In phylogenetic analysis, *Na*Gal-1 and galectin-1 of other fishes formed one cluster. The structural similarity and conservative evolutionary relationship of *Na*Gal-1 protein strongly prove that *Na*Gal-1 is a member of *galectin-1*.

The results of qRT-PCR showed that the mRNA expressions of *NaGal-1* were ubiquitously distributed in various tissues, especially in swim-bladder and gill. It has been reported that galectin can promote angiogenesis, and there are a large number of blood vessels on the surface of gill and swim-bladder, so we can speculate that *Na*Gal-1 may be involved in angiogenesis of swim-bladder and gill surface. In addition, gills are the first line of defense against pathogens in fish, and they contain a large number of immune cells, such as lymphocytes, neutrophils, and macrophages, which can secrete large amounts of galectin-1 [21]. The expression of *Na*Gal-1 in brain significantly increased after infection with *V. harveyi*. Microglias in the brain are the first line of immune defense of the central nervous system, and are activated during systemic infection with bacteria, and then producing soluble pro-inflammatory mediators that can target the infectious agents [22]. In addition, galectins has been shown to have the potential to regulate inflammatory response [23,24]. To sum up, we speculated that under the infection of *V. harveyi*, the microglial cells in the brain may secrete a large amount of *NaGal-1* to participate in the inflammatory response, and the specific immune functions of *NaGal-1* in brain need to be further studied.

Expression of *Na*Gal-1-EGFP in HEK 293T cells was distributed in the cytoplasm and nucleus, and the result was consistent with expression of Galectin 1-like 2 (CiGal1-L2) in *Ctenopharyngodon idella* studied by Zhu, D et al. Interestingly, the CiGal1-L2 protein showed a tendency of nuclear translocation after LPS and Poly I:C treatment [25], which may be related to the fact that galectin-1 is a redundant pre-mRNA splicing factor [26]. *Na*Gal-1 protein is lack of signal peptide and transmembrane region, and therefore it cannot be secreted according to the classical pathway. Galectins were synthesized in the cytoplasm and widely presented in immune cells. Taken together, *Na*Gal-1-EGFP protein was a nonclassical secretory protein, which can be secreted through non-classical pathway to participate in a variety of immune response.

Immune function analysis was performed on recombinant *Na*Gal-1 protein. The *Na*Gal-1 protein could agglutinate RBCs from rabbit, *L. crocea*, and *N. albiflora,* the agglutination activity of the *Na*Gal-1 protein toward *N. albiflora* RBCs was inhibited by PGN, LPS, lactose, and D-galactose. In addition, the *Na*Gal-1 protein agglutinated and killed some targeted gram-negative bacteria. Our previous studies have shown that a galectin-3 protein of *N. albiflora* agglutinated some gram-negative bacteria including *P. plecoglossicida*, *V. parahemolyticus*, *V. harveyi*, and *A. hydrophila*, and disturbed the cell wall of *V. parahemolyticus* and *V. harveyi* [27]. The galectin-2 protein of *O. niloticus* could agglutinate both gram-negative bacteria (*E. coli*, *V. harveyi*, *V. Parahemolyticus*, and *V. alginolyticus*) and gram-positive bacteria (*Streptococcus agalactiae*, *Streptococcus iniae*, *Staphylococcus pasteuri*, and *Lactococcus garviea*) [13]. The recombinant galectin-1 protein of *Channa striatus* could agglutinate the mouse RBCs, and this activity was inhibited by D-galactose, D-glucose and D-fructose at certain concentrations, and it also agglutinated only gram-negative bacteria including *E. coli*, *A. hydrophila*, *V. vulnificus*, *V. alginolyticus*, and *P. aeruginosa* [28]. Recombinant galectin-1 protein of *Epinephelus coioides* made chicken RBCs aggregation and agglutinated gram negative bacteria (*E. coli* DH5α, *E. coli* BL21 (DE3) and gram positive bacteria (*Staphylococcus aureus*, *Micrococcus lysodeikticus*) [29]. Additionally, A galectin-1 protein from *Apostichopus japonicus* could bind various PAMPs including D-galactose and LPS, and exhibited the highest affinity to D-galactose [30]. Generally, the basic unit recognized by all galectins is Gal-β(1-4)-GlcNAc [20]. Some galectins have also shown interaction with bacterial LPSs by binding to β-galactoside residues of LPS [31]. Therefore, we speculated that sugar binding motifs (H-NPR and W--E-R) of *Na*Gal-1 protein directly bound to *β*-galactoside-containing polysaccharide chain of LPS. Davicino, R.C. et al. reported that galectin-1 served as receptor for an extracellular pathogen capable of adhering to lipophosphoglycan structures rich in galactose and N-acetylglucosamine [4], while N-acetylglucosamine was an amino sugar that formed the lattice structure of PGN. We thus speculated that *Na*Gal-1 protein directly bound to N-acetylglucosamine of PGN, and galectins recognize and kill bacteria through distinct antigenic determinant (bacteria generate a wide variety of glycan-based antigenic structures) [32,33]. Accumulating evidence indicates that galectins can not only recognize β-galactoside-containing polysaccharide, but also bind to glycans (such as: LPS and PGN) that are specific to pathogens, thus initiate immune responses and clearance pathogens, most galectins bind directly to glycans on the surface of specific cells through traditional ligand–receptor interactions [4,34]. Galectins are members of PRRs in the innate immune response against pathogen infection, and they can play a direct and indirect antibacterial effect when they bind to pathogen surface [35]. *Na*Gal-1 protein as a PRR may participate in immune response of *N. albiflora* by recognizing and interacting PAMPs on specific pathogenic bacteria.

## 4. Materials and Methods

### 4.1. Experimental Fish and Bacterial Infection Experiment

Young *N. albiflora* (3.28 ± 1.71 g) collected from Ningde City (Fujian Province, China). The cultivation and infection of *N. albiflora* with *V. harveyi* were carried out according to a previously described protocol [36]. In order to examine the distribution of *NaGal-1* mRNA, twelve tissues including brain, heart, swim-bladder, kidney, liver, skin, gill, stomach, head-kidney, spleen, intestine, and muscle were collected from six healthy *N. albiflora*. In addition, the *V. harveyi* infection was carried out for studying the innate immune response of *Na*Gal-1 during the *V. harveyi* infection. After 6 h, 12 h, 24 h, 48 h, 72 h, and 96 h of the *V. harveyi* infection, samples from brain, head-kidney, liver, and spleen were collected and all the collected tissues were frozen by liquid nitrogen and stored in −80 °C for further analysis.

### 4.2. RNA Isolation and cDNA Synthesis

Total RNAs was extracted from the collected samples using TransZol Up Plus RNA Kit (TransGen Biotech, Beijing, China) in accordance with the protocol of manufacture. cDNA synthesis was performed from total RNA with GoScript™ Reverse Transcription System (Promega, Madison, WI, USA) according to manufacturer’s instructions.

### 4.3. Cloning of NaGal-1 Gene

A pair of specific primers (*Gal-1*-F and *Gal-1*-R, Table 2) were designed with *EcoR* I and *Xho* I restriction sites for the open reading frame (ORF) amplification. PCR was carried out as follows: initial denaturation step at 95 °C for 15 s, 35 cycles of amplification (95 °C for 15 s, 58 °C for 15 s, and 72 °C for 30 s), followed by a final extension step at 72 °C for 5 min. After gel-purification, *NaGal-1* ORF was recombined into *pET-32a* vector with ClonExpress^®®^ II One Step Cloning Kit (Vazyme Biotech, China), and the clones were selected and sequenced in Sanger sequencing in BioSune (Shanghai, China).

### 4.4. Bioinformatics Analysis

The homologous analysis of amino acid sequences was conducted using the Clustal Omega (https://www.ebi.ac.uk/Tools/msa/clustalo/ (accessed on 15 August 2020)). The conserved domains of *Na*Gal-1 were identified using SMART (http://smart.embl-heidelberg.de/smart/set_mode.cgi?NORMAL=1 (accessed on 25 August 2020)). The tertiary structure and signal peptides were predicted by SWISS-MODEL (http://swissmodel.expasy.org/interactive (accessed on 7 October 2020)) and SignaIP program (http://www.cbs.dtu.dk/services/SignalP/ (accessed on 27 October 2020)), respectively. Phylogenetic tree was constructed by maximum-likelihood method at MEGA 6.06.

### 4.5. Tissue Distribution of NaGal-1 and Temporal Expression Pattern Post Infection

qRT-PCR was employed to detect the tissue-specific expression of *NaGal-1* and the time-course expression profiles after *V. harveyi* infection. Two gene specific primers (*qGal-1*-F and *qGal-1*-R, Table 2) were used for *Na*Gal-1 fragment amplification. *β-actin* was used as reference gene (*β-actin*-F and *β-actin*-R, Table 2). The qRT-PCR was performed using ChamQTM Universal SYBR^®®^ qPCR Master Mix (Vazyme Biotech, Nanjing, China) on an Applied Biosystems QuantStudio 6&7 Real-time PCR System (Application Biosysterms, Waltham, MA, USA). The cycle profiles were pre-denaturized at 95 °C for 30 s, followed by 40 cycles at 95 °C for 10 s and annealing at 60 °C for 30 s. Each sample was tested in triplicates. The 2^−ΔΔCt^ method was used to analyze the expression level.

### 4.6. Expression of NaGal-1 in HEK 293T Cells

For the purpose of studying expression of *Na*Gal-1 protein in HEK 293T Cells, the ORF sequence of *NaGal-1* was cloned into *pEGFP-N1* vector using ClonExpress^®®^ II One Step Cloning Kit (Vazyme Biotech, Nanjing, China) with its specific primers (*sGal-1*-F and *sGal-1*-R, Table 2). The product was named as *NaGal-1-EGFP*. Human embryonic kidney 293T (HEK 293T) cells were seeded into 12-well plates and cultured for 24 h in DMEM with 10% fetal bovine serum and 1% penicillin-streptomycin solution (Biological Industries, Cromwell, CO, USA) under a humidified condition with 5% CO_2_ at 37 °C. Afterwards, 1 μg of plasmids of *NaGal-1-EGFP* and *pEGFP-N1* (control group) were respectively transfected into the cells using Lipo 8000™ Transfection Reagent (Beyotime, Shanghai, China).

Cells were harvested and lysed with cell lysis buffer (Beyotime, Shanghai, China), followed by Western blotting to verify *Na*Gal-1-EGFP and EGFP protein. The Western blotting procedure was as follows: the protein samples were processed by 12% SDS-PAGE and transferred to polyvinylidene difluoride membrane. The membrane was blocked with TBST buffer containing 5% BSA (TRIS-Cl 20 mL, NaCl 150 mM, pH 8.0, TEN-20 0.1%, TRIS-20 buffer) for 2 h at room temperature. The cells were then incubated with GFP Rabbit Monoclonal Antibody (diluted 1:1000, Beyotime, Shanghai, China) overnight at 4 °C. After washing with TBST 5 times (5 min each time), the samples were incubated with HRP labeled goat anti-rabbit IgG (diluted 1:1000, Shanghai Biyuntai) for 2 h, and then washed with TBST 7 times (5 min each time). Membranes were detected by BeyoECL Plus (Beyotime, Shanghai, China), and digital images were acquired by ImageQuant LAS 4000 (GE Healthcare, Inc., Chicago, IL, USA) [37].

Additionally, at 24 h after transfection, cells were fixed with 4 % (*v*/*v*) paraformaldehyde, permeabilized with 0.2% Triton X-100, and stained with 4,6-diamidino-2-phenylindole (DAPI, Beyotime, Shanghai, China). Using the confocal fluorescence microscope Leica TCS SP8 system (lycra, Niederfrohna, Germany) to observe expression in HEK 293T cells of proteins.

### 4.7. Prokaryotic Expression and Purification of NaGal-1 Protein

The prokaryotic expression was performed in *E. coli* BL21 (DE3). When OD_600_ reached 0.6–0.8, isopropyl *β*-D-thiogalactoside (IPTG) was added to a final concentration of 0.01 mM and induced with shaking at 200 rpm at 37 °C for 6 h. The bacterial solution was collected (4 °C and 7000× *g* centrifuging for 5 min) and washed three times with imidazole buffer (20 mM imidazole, 10 mM Na_2_HPO_4_, 140 mM NaCl, 1.8 mM KH_2_PO_4_, 2.7 mM KCl). Then resuspension was carried out with 20 mL of imidazole buffer (20 mM) and ultrasonically lysed was performed on ice at 50% power for 30 min. The supernatant was then collected (4 °C, centrifuged at 12,000× *g* for 10 min), incubated with Ni SepharoseTM 6 Fast Flow (GE Healthcare), and then eluted with buffer containing 500 mM imidazole. Trx protein expressed by *pET-32a* vector was used as a negative control. The purity of protein was analyzed by 12% SDS-PAGE. The samples of protein were freeze-dried after dialysis and stored at −80 °C.

### 4.8. Hemagglutination and Sugar Inhibition Assays

The hemagglutination assay was performed following the reported method. Briefly, red blood cells (RBCs) sample from rabbit, *L. crocea*, and *N. albiflora* were collected and washed three times in Tris buffered saline (TBS buffer: 20 mM Tris, 150 mM NaCl, pH 8.0) and TBS+Ca^2+^ buffer (20 mM Tris, 150 mM NaCl, 10 mM CaCl_2_, pH 8.0), and the cell number was adjusted with TBS buffer to 2% suspension. 

Equal volumes of two-fold serially diluted *Na*Gal-1 (50 μL) dissolved in TBS buffer and TBS+Ca^2+^ buffer, were incubated with RBCs suspension (50 μL) in a micro-titer U plate at room temperature for 1 h. GST protein in TBS buffer and TBS+Ca^2+^ buffer was chosen as negative control. The hemagglutination of RBCs was observed under the TCS SP8 system (Leica, Wetzlar, Germany). 

Sugar inhibition of *Na*Gal-1 was determined based on the inhibition of hemagglutination. Twelve sugars including PGN, LPS, D-mannose, D-trehalose dihydrate, D-galactose, lactose, D-fructose, D-glucose, D-maltose monohydrate, D-mannitol, D-raffinose pentahydrate, and sucrose were used as substrates in this study. Two-fold serial dilutions (25 μL), starting at 1.25 mg/mL of sugars with a series of dilution were mixed with protein (25 μL), and incubated at room temperature for 30 min. Then, 50 μL of 2% *N. albiflora* RBCs suspension was added into each mixture and incubated at room temperature for 1 h. The degree of inhibition of hemagglutination was observed under the microscope after an hour. Assays were performed in triplicates.

### 4.9. Bacterial Agglutination and Live/Dead Tests

To investigate the bacterial agglutination activity of *Na*Gal-1, two-fold serial dilutions (50 μL) started at 3 mg/mL of *Na*Gal-1 protein was used. As the bacteria were cultured to the logarithmic stage, they were collected by centrifugation at 3000× *g* for 10 min, washed three times and re-suspended in 0.85% NaCl buffer at 2 × 10^8^ CFU/mL. The *Na*Gal-1 protein and bacterial suspension were incubated together at room temperature for 1 h and stained with Live/Dead^®®^BacLightTMBacterial Viability Kit (Invitrogen, Eugene, OH, USA). SYTO 9 stain (6 µM) and propidium iodide (30 µM) were mixed with same volume of the bacterial suspensions, and then incubated in the dark for 15 min, the SYTO9 stain labels bacteria with both intact and damaged cell wall and the bacteria show green fluorescence, while the propidium iodide (PI) stain only labels bacteria with damaged cell wall and the dead bacteria show red fluorescence. The images were acquired by the Leica TCS SP8 system (Leica, Wetzlar, Germany).

### 4.10. Statistical Analysis

Statistical analysis was performed by one-way ANOVA and LSD multiple comparison test using the SPSS 20.0 (IBM, Armonk, NY, USA), for evaluating the significant differences among samples at the 5% significant level. All experiments replicated three times.

## 5. Conclusions

In this study, we identified and characterized a galectin-1 from *N. albiflora* (named as *NaGal-1*). *NaGal-1* was ubiquitously distributed in all the detected tissues. After infection with *V. harveyi*, the expression of *NaGal-1* in the brain increased significantly, suggesting that *Na*Gal-1 protein may be involved the specific immune functions of brain of *N. albiflora*. *Na*Gal-1 protein was a nonclassical secretory protein and its expression in HEK 293T cells was distributed in the cytoplasm and nucleus. The purified Trx-*Na*Gal-1 protein through prokaryotic expression could agglutinate RBCs from Rabbit, *L. crocea* and *N. albiflora* in a Ca^2+^-independent manner, and the activity of hemagglutination was inhibited by LPS, PGN, lactose, and D-galactose. Trx-*Na*Gal-1 protein could agglutinate bacteria including *E. tarda*, *E. coli*, *P. phosphoreum*, *A. hydrophila*, *P. aeruginosa*, and *A. veronii*, but did not agglutinate bacteria including *P. plecoglossicida*, *V. parahemolyticus*, *V. harveyi*, etc., indicating its function of microbial recognition. Moreover, *Na*Gal-1 killed the agglutinated bacteria. These results suggested that *Na*Gal-1 protein might play an important role in the innate immunity of *N. albiflora*.

## Figures and Tables

**Figure 1 ijms-24-03298-f001:**
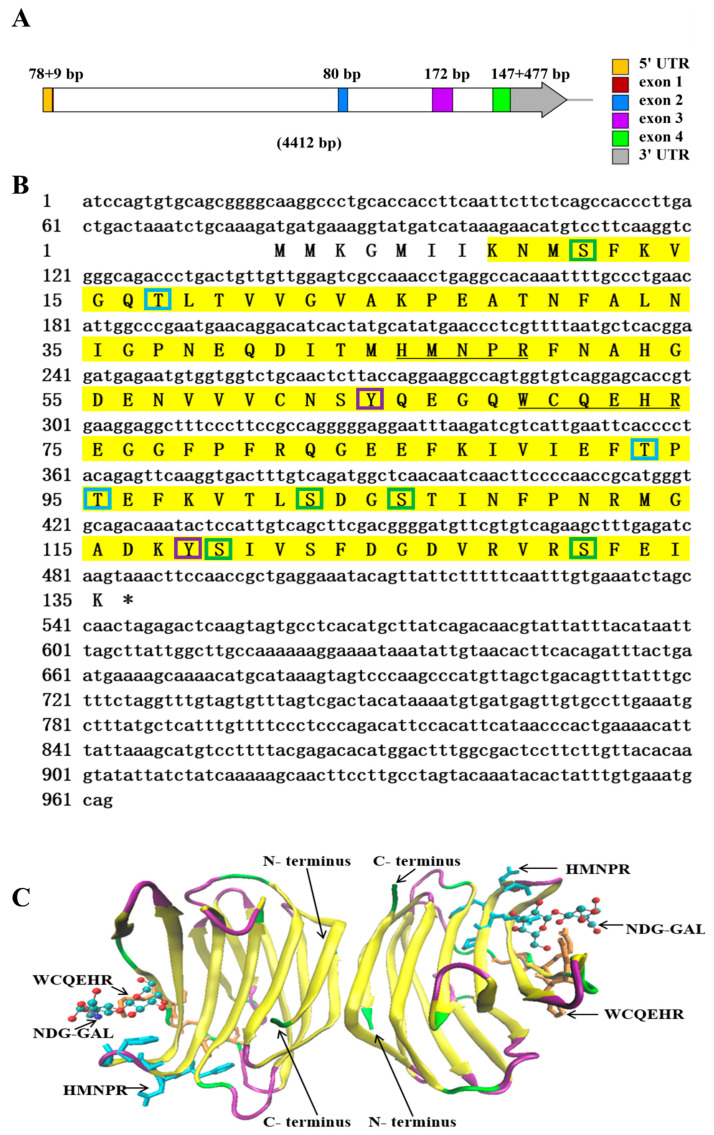
The sequence and structure analysis of *NaGal-1*. (**A**) The diagram of *NaGal-1* gene structure in *N. albiflora*. The squares in different color within the arrow area represent the different exons of *NaGal-1* gene and the white squares represent introns. The lengths of exons are shown on the top. (**B**) The stop codons were indicated with asterisk (*), the CRD was highlighted in yellow shade, the sugar binding motifs H-NPR and W--E-R were underlined, the green boxes indicate Ser phosphorylation sites, the blue boxes indicate Thr phosphorylation sites, the purple boxes indicate Tyr phosphorylation sites. (**C**) Tertiary structure of *Na*Gal-1 protein is a homodimer, two conserved *β*-galactoside binding motifs (HFNPR and WGPEER), C-terminus and N-terminus, and ligands (NDG-GAL) are labeled with arrows. *β*-sheets, random coils, and *β*-turns are shown in yellow, green, and purple, respectively.

**Figure 2 ijms-24-03298-f002:**
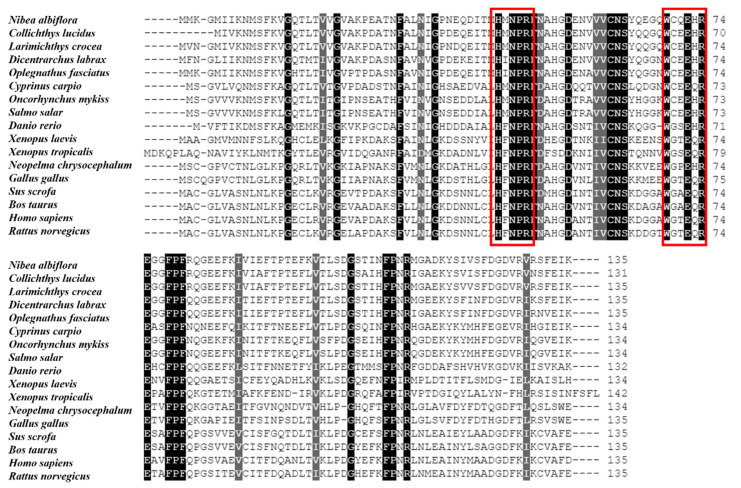
Multiple alignment of NaGal-1 amino acid sequences. The red boxes indicate the sugar binding motifs (H-NPR and W--E-R), and the black shadows indicate identical amino acid.

**Figure 3 ijms-24-03298-f003:**
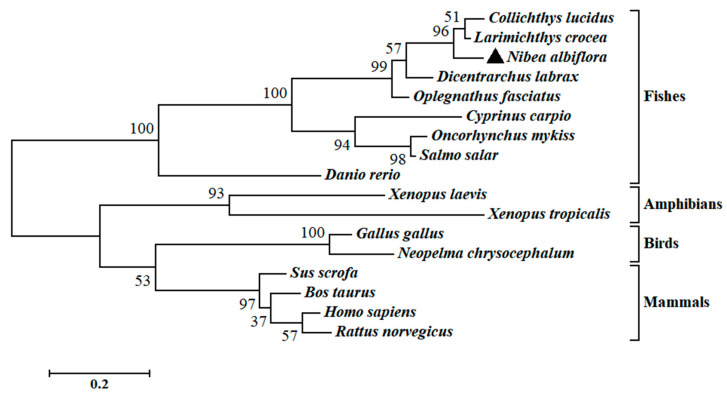
Phylogenetic analysis of *Na*Gal-1. *N. albiflora* is black triangled, the numbers at the nodes indicate the bootstrap confidence values (100%) of 1000 replicates, the scale bar (0.2) indicates the genetic distance, the GenBank accession numbers of amino acid sequences used are listed in Table 1.

**Figure 4 ijms-24-03298-f004:**
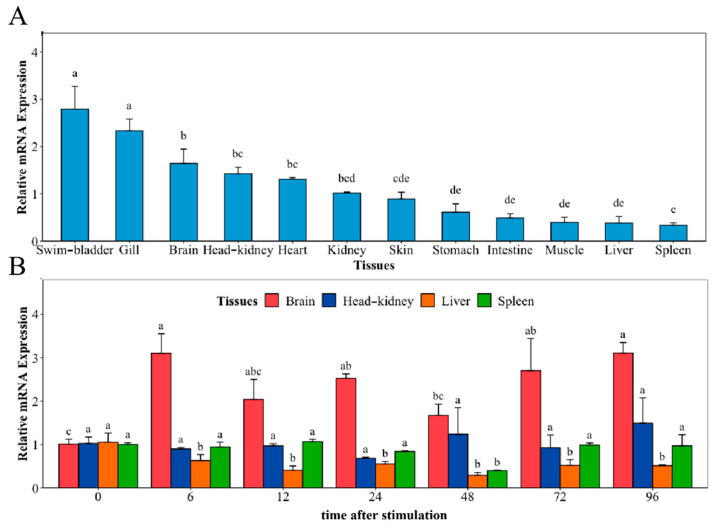
Relative mRNA expression of *NaGal-1* (**A**) in twelve tissues of *N. albiflora*, and (**B**) after *V. harveyi* infection in brain, head kidney, liver, and spleen. The letters a, b, c, d, and e denote statistical significance (*p* < 0.05). The same letter indicates no significant differences (*p* > 0.05), adjacent letters indicate significant differences (*p* < 0.05), separated letters indicate a significant difference (*p* < 0.01).

**Figure 5 ijms-24-03298-f005:**
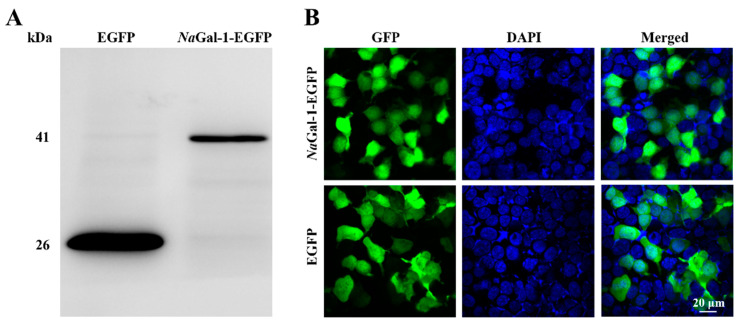
Expression of *Na*Gal-1 in HEK 293T Cells. (**A**) Western blot of *Na*Gal-1-EGFP (41.26 kDa) compared with control EGFP (26 kDa), and (**B**) Expression of *Na*Gal-1-EGFP in HEK 293T cells was found in both cytoplasm and nucleus observed under the confocal fluorescence microscopy.

**Figure 6 ijms-24-03298-f006:**
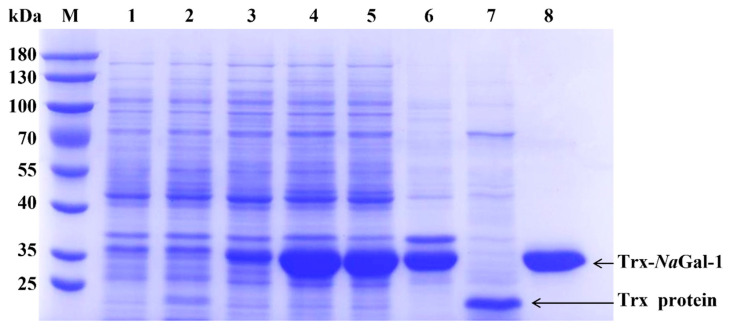
SDS-PAGE analysis of *Na*Gal-1 fusion protein. Protein marker (lane M); proteins of *E. coli* BL21 containing control vector *pET-32a*, non-induced (lane 1), induced by IPTG (lane 2), and purified Trx protein (lane 7); proteins of *E. coli* BL21 containing *pET-32a-NaGal-1*, non-induced (lane 3), induced (lane 4); proteins of *E. coli* BL21 containing pET-32a-*Na*Gal-1 after IPTG induction for 6 h and ultrasonically lysed (supernatant) (lane 5), and precipitation (lane 6) and purified Trx-*Na*Gal-1 protein (lane 8).

**Figure 7 ijms-24-03298-f007:**
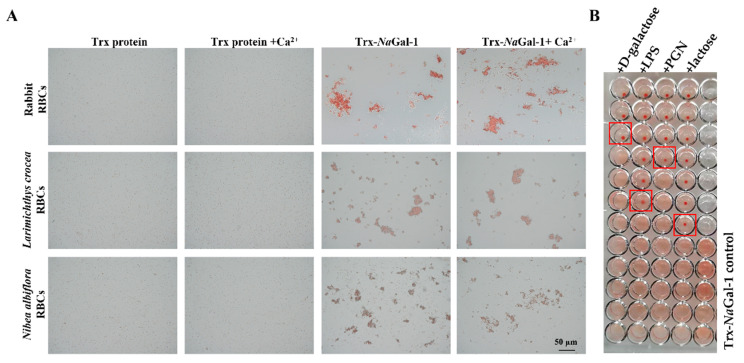
Hemagglutination and sugar inhibition assay of Trx-*Na*Gal-1 protein. (**A**) Hemagglutination of Trx-*Na*Gal-1 protein toward RBCs from rabbit, *L. crocea*, and *N. albiflora*, the Trx protein groups were as negative control. (**B**) The red boxes indicate the minimum inhibitory concentration of sugars (From top to bottom are corresponding to high-to-low concentrations of competitive sugars: the initial concentration of PGN and LPS are 12.22 ng/mL, the initial concentration of D-galactose and lactose are 39.06 µg/mL, and then two-fold serial dilutions are made from top to bottom), the minimum inhibitory concentrations of Trx-*Na*Gal-1 protein to LPS, PGN, lactose, and D-galactose were 0.38 ng/mL, 1.52 ng/mL, 2.44 µg/mL, and 19.53 µg/mL, respectively. This assay was performed in triplicates.

**Figure 8 ijms-24-03298-f008:**
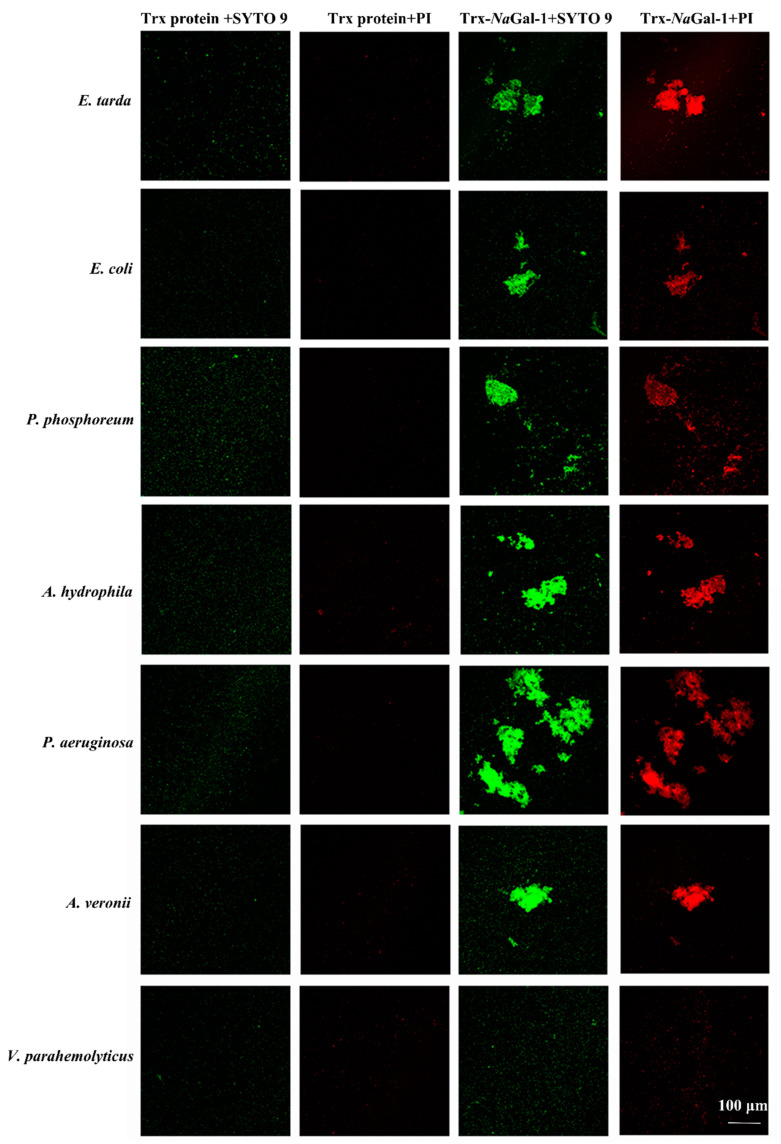
Bacterial agglutination and antibacterial activity of Trx-*Na*Gal-1 protein. Agglutinated and dead bacteria: *E. tarda*, *E. coli*, *P. phosphoreum*, *A. hydrophila*, *P. aeruginosa*, and *A. veronii*, and non-agglutinated bacteria representative: *V. parahemolyticus*. All the bacteria stained by SYTO 9 were in green and dead bacteria stained by propidium iodide (PI) were in red.

**Table 1 ijms-24-03298-t001:** Amino acid sequence identity between *Na*Gal-1 and Gal-1 of other vertebrates.

GenBank Accession Numbers	Species	Identity (%)
XP_010736086.1	*Larimichthys crocea*	90.37
TKS91807.1	*Collichthys lucidus*	85.93
ADV35589.1	*Oplegnathus fasciatus*	80.74
ACF77003.1	*Dicentrarchus labrax*	80.00
NP_001134631.1	*Salmo salar*	61.48
ACO07656.1	*Oncorhynchus mykiss*	60.74
XP_018930790.1	*Cyprinus carpio*	60.00
XP_005172121.1	*Danio rerio*	47.41
AAK11514.1	*Xenopus laevis*	39.26
NP_002296.1	*Homo sapiens*	37.78
NP_786976.1	*Bos taurus*	37.04
XP_027535424.1	*Neopelma chrysocephalum*	37.04
NP_063969.1	*Rattus norvegicus*	37.04
NP_990826.1	*Gallus gallus*	36.76
NP_001001867.1	*Sus scrofa*	36.30
XP_017949150.2	*Xenopus tropicalis*	31.25

**Table 2 ijms-24-03298-t002:** Primers used in this study.

Primer Name	Sequence (5′-3′)	Purpose
*Gal-1*-F	GCTGATATCGGATCCGAATTCATGATGAAAGGTATGATCATAAAGAACA	ORF amplification
*Gal-1*-R	GTGGTGGTGGTGGTGCTCGAGTTACTTGATCTCAAAGCTTCTGACACG	
*qGal-1*-F	GAGGAATTTAAGATCGTCATTGAAT	qRT-PCR analysis
*qGal-1*-R	CTCAAAGCTTCTGACACGAACATCC	
*β-actin*-F	TTATGAAGGCTATGCCCTGCC	
*β-actin*-R	TGAAGGAGTAGCCACGCTCTGT	
*sGal-1*-F	CTACCGGACTCAGATCTCGAGATGATGAAAGGTATGATCATAAAGAACA	Subcellular localization
*sGal-1*-R	GTACCGTCGACTGCAGAATTCCCTTGATCTCAAAGCTTCTGACACG	

Note: *EcoR* I (GAATTC) and *Xho* I (CTCGAG) enzyme restriction sites are underlined.

## Data Availability

Not applicable.

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
