# Peer review of "Molecular Cloning and Functional Characterization of Galectin-1 in Yellow Drum (*Nibea albiflora*)"

_ijms, 2023, doi:10.3390/ijms24043298_

Round 1

Reviewer 1 Report

The authors reported the characterization of Galectin-1 from a fish, yellow drum (Nibea albiflora) named YdGal-1. They found the YdGal-1 is homology of Galectin-1 in other animals. YdGal-1 is expressed in multi-organs and could be upregulated by Vibrio harveyi attack. Recombinant YdGal-1 produced in E coli could agglutinate red blood cells from rabbit, Larimichthys crocea, and N. albiflora. In addition, recombinant YdGal-1 could neutralize some gram-negative bacteria. Their findings showed a possible role of YdGal-1 in the innate immunity of this fish to defense the pathogens. It is valuable to publish this paper benefiting the scientific community of fish. However, the manuscript needs some improvement before publication.

1) The title is “ Antibacterial Galectin-1 identified from yellow drum”. however, how the Galectin-1 of yellow drum was identified was not reported.

2) The Galectin-1 of yellow drum (Nibea albiflora) is better to name as NaGal-1 instead of YdGal-1, because using the Latin name is better than the common name.

3) The language need be improved. Some typos are in the text, and some sentences are difficult to understand. Examples:

Line 13, “It consists…”. What is it?

Line 32-37, “Galectins…a short peptide.”

Line 50, “like other fish Galectins”. What are the other fish Galectins?

Line 59-60, the last sentence is a repeat of the previous sentence “can participate in the immune response of fish”.

Line 99, Collichthys lucidus should be in Italic.

Line 120, “lever”.

Figure 4, “time of Post challenge”.

4) Figure 1, A, What is the numbers’ meaning and where iis it from in the axis? 5 and 3-UTR should be 5’ and 3’-UTR.

5) Line 95, “phylogenetic tree analysis” must be “phylogenetic analysis”.

6) Figure 4, The meaning of different letters above the columns should be noted in the legend.

7) Line 130, “Subcellular localization of YdGal-1” should be changed as “expression of NaGal-1 in HEK 293T Cells”. The original title made a confusion about expression of NaGal-1 in the cells of yellow drum.

8) Line 163-172. The results were missing. The images or the diagrams or the tables should present in the manuscript. Only some sentences are not enough as evidence.

9) Line 177-181. It is the description of the method and it is better to move into the Methods or in the legend of Figure 8.

10) Line 235-260, the last paragraph in the Discussion. The authors described the studies about Galectin-1 and other Galectins in the fish. However, it is a data description instead of discussion, and it can be described in brief.

Author Response

Dear Editor and Reviewer 1,

Thank you very much for your careful work and valuable comments. Our manuscript “Antibacterial Galectin-1 identified from yellow drum(Nibea albiflora)” (Manuscript ID ijms-2115815) has been revised according to the reviewers’ comments. Those comments are very helpful for revising and improving our paper. We have studied the comments carefully and made corrections. The main corrections are in the manuscript and the responses to the reviewer’ comments are as follows (the replies are highlighted in blue).

1) The title is “Antibacterial Galectin-1 identified from yellow drum”. however, how the Galectin-1 of yellow drum was identified was not reported.

Response: the title has been changed into “Molecular cloning and functional characterization of Galectin-1 in yellow drumNibea albiflora”.

2)The Galectin-1 of yellow drum (Nibea albiflora) is better to name as NaGal-1 instead of YdGal-1, because using the Latin name is better than the common name.

Response: We thank the reviewer’s helpful suggestion. We have used “NaGal-1" instead of "YdGal-1".

3) The language need be improved. Some typos are in the text, and some sentences are difficult to understand. Examples:

Response: We have studied the manuscript carefully and made corrections.

  • Line 13, “It consists…”. What is it?

In the revised manuscript, it has been changed into “Tertiary structure of NaGal-1 protein consists of homodimers and each dimer has one carbohydrate recognition domain.”

  • Line 32-37, “Galectins…a short peptide.”

It has been changed in the revised manuscript.

  • Line 50, “like other fish Galectins”. What are the other fish Galectins?

It has been changed into “like other galectins from fish (such as: galectin-3, -8, -9)”.

  • Line 59-60, the last sentence is a repeat of the previous sentence “can participate in the immune response of fish”.

The last sentence has been deleted in the revised manuscript.

  • Line 99, Collichthys lucidus should be in Italic.

Collichthys lucidus has been in italic in the revised manuscript .

  • Line 120, “lever”.

The word “level” has been corrected in the revised manuscript.

  • Figure 4, “time of Post challenge”.

Figure 4 marking of abscissa has been changed into "time after stimulation".

4) Figure 1, A, What is the numbers’ meaning and where is it from in the axis? 5 and 3-UTR should be 5’ and 3’-UTR.

Response: The lengths of exons are shown on the top in figure 1. And 5’ and 3’-UTR have been corrected in the revised manuscript.

5) Line 95, “phylogenetic tree analysis” must be “phylogenetic analysis”.

Response: we have used “phylogenetic analysis" instead of "phylogenetic tree analysis ".

6) Figure 4, The meaning of different letters above the columns should be noted in the legend.

Response: In the revised manuscript, we have added some interpretation of figure 4:”The letters a, b, c and d denote statistical significance (p < 0.05). The same letter indicates no sig-nificant differencesp > 0.05),adjacent letters indicate significant differencesp < 0.05,separated letters indicate a significant differencep <0.01.”

7) Line 130, “Subcellular localization of YdGal-1” should be changed as “expression of NaGal-1 in HEK 293T Cells”. The original title made a confusion about expression of NaGal-1 in the cells of yellow drum.

Response: We agree with the views and suggestions of the reviewer, so we changed into "expression of NaGal-1 in HEK 293T Cells" in Line 130.

8) Line 163-172. The results were missing. The images or the diagrams or the tables should present in the manuscript. Only some sentences are not enough as evidence.

Response:As suggested, figure has been added accordingly.

9) Line 177-181. It is the description of the method and it is better to move into the Methods or in the legend of Figure 8.

Response: As suggested, this section has been moved into the method.

10) Line 235-260, the last paragraph in the Discussion. The authors described the studies about Galectin-1 and other Galectins in the fish. However, it is a data description instead of discussion, and it can be described

Response: As suggested, the last paragraph in the discussion has been revised in brief accordingly.

Reviewer 2 Report

The manuscript, titled “Antibacterial Galectin-1 identified from yellow drumNibea albiflora” by Wu et al., describes the identification of YdGal-1 a galectin from N. albiflora, and its presumed structure, expression pattern, localization and biochemical property. This study might be important to researchers of glycobiology and immunology. However, this manuscript raises multiple concerns that should be addressed.

#1: Primary structure and bioinformatic analysis

In line 198, it is described that “In this study, we identified a galectin-1 (named YdGal-1) from N. albiflora and analyzed its sequence by bioinformatics”. The result is presumed to be presented in Figure 1. However, the method to identify the YdGal-1 and the accession number of its sequence could not be found in this manuscript. Such information is essential.

              As to the presumed ternary structure of YdGal-1, it is described that “Tertiary structure of YdGal-1 protein is a two-fold symmetric dimer (Figure 1C), and the two monomers interact with each other by an α-helix” in lines 81-83. Meanwhile, a-helix was not found in the structure of YdGal-1, and in the legend to figure 1, it is described that “β-sheets, random coils, and β-turns are shown in yellow, green, and purple, respectively”. The statement that YdGal-1 does not have an α-helix but forms a dimer through α-helix is contradictory.

#2: Sugar inhibition assay

Sugar-binding capacity is one of the important biochemical properties of galectins. Galectins are generally known as galactose-binding lectins, and there are not many reports of their binding to LPS or PGN. In this respect, the present finding is unique. However, the data is only mentioned in the text (lines 163-172) and not presented. The data should be presented as a figure. In addition, a more detailed discussion of the molecular mechanism by which YdGal-1 binds to LPS and PGN would be desirable.

#3: Manuscript flow, expression, and (potential) errors

Since the manuscript flow and expression could be improved and there might be multiple errors or unnatural descriptions in this version, it is strongly recommended to check through the manuscript carefully and make the appropriate correction.

Several examples are below:

Line 36-:

Tandem-repeat galectins have two CRDs. They are similar, but not identical. In this regard, the description ‘homologous’ may lead to misleading

Line 120:

‘lever’ should be ‘level’.

Line 152-:

The legend to Figure 6 is difficult to understand. Particularly, the difference between “induced supernatant (lane5)” and “induced precipitation (lane 6)”. In addition, the detailed method for sample preparation could not be found in Materials and Methods.

Line 215:

The expression "nuclear macrophage" is not commonly used. This might be an error.

Author Response

Dear Editor and Reviewer 2,

Thank you very much for your careful work and valuable comments. Our manuscript “Antibacterial Galectin-1 identified from yellow drum(Nibea albiflora)” (Manuscript ID ijms-2115815) has been revised according to the reviewers’ comments. Those comments are very helpful for revising and improving our paper. We have studied the comments carefully and made corrections. The main corrections are in the manuscript and the responses to the reviewer’ comments are as follows (the replies are highlighted in blue).

#1: Primary structure and bioinformatic analysis

In line 198, it is described that “In this study, we identified a galectin-1 (named YdGal-1) from N. albiflora and analyzed its sequence by bioinformatics”. The result is presumed to be presented in Figure 1. However, the method to identify the YdGal-1 and the accession number of its sequence could not be found in this manuscript. Such information is essential.

Response: In this study, through sequence analysis, homology comparison and phylogenetic analysis, Gal1 from yellow drumNibea albiflorabelongs to the family of Galectin 1 (GemBank No.OQ139528).

As to the presumed ternary structure of YdGal-1, it is described that “Tertiary structure of YdGal-1 protein is a two-fold symmetric dimer (Figure 1C), and the two monomers interact with each other by an α-helix” in lines 81-83. Meanwhile, a-helix was not found in the structure of YdGal-1, and in the legend to figure 1, it is described that “β-sheets, random coils, and β-turns are shown in yellow, green, and purple, respectively”. The statement that YdGal-1 does not have an α-helix but forms a dimer through α-helix is contradictory.

Response: We thank the reviewer’s question. We have been revised accordingly. NaGal-1 protein is homodimers composed of subunits of 135 amino acids, have not an α-helix.

#2: Sugar inhibition assay

Sugar-binding capacity is one of the important biochemical properties of galectins. Galectins are generally known as galactose-binding lectins, and there are not many reports of their binding to LPS or PGN. In this respect, the present finding is unique. However, the data is only mentioned in the text (lines 163-172) and not presented. The data should be presented as a figure. In addition, a more detailed discussion of the molecular mechanism by which YdGal-1 binds to LPS and PGN would be desirable.

Response: We thank the reviewer’s useful suggestion. As suggested, sugar inhibition assay has been provided as Figure 7B.

Molecular mechanism binds to LPS and PGN: most galectins bind directly to LPS and PGN through traditional ligand-receptor interactions, and sugar binding motifs were the key factor to perform function of galectin, H-NPR and W--E-R were directly involved in carbohydrate binding.

#3: Manuscript flow, expression, and (potential) errors

Since the manuscript flow and expression could be improved and there might be multiple errors or unnatural descriptions in this version, it is strongly recommended to check through the manuscript carefully and make the appropriate correction.

Several examples are below:

Line 36-: Tandem-repeat galectins have two CRDs. They are similar, but not identical. In this regard, the description ‘homologous’ may lead to misleading.

Response: homologous has been changed into corresponding.

Line 120: ‘lever’ should be ‘level’.

Response: The word “level” has been corrected in the revised manuscript.

Line 152-: The legend to Figure 6 is difficult to understand. Particularly, the difference between “induced supernatant (lane5)” and “induced precipitation (lane 6)”. In addition, the detailed method for sample preparation could not be found in Materials and Methods.

Response: We thank the reviewer’s helpful suggestion. As suggested, the legend to Figure 6 has been revised accordingly. And the detailed method for sample preparation has been added in section 4.7

Line 215: The expression "nuclear macrophage" is not commonly used. This might be an error.

Response: We thank the reviewer’s helpful suggestion, we have changed it as "macrophage" in Line 215.

Round 2

Reviewer 2 Report

The manuscript, titled “Antibacterial Galectin-1 identified from yellow drumNibea albiflora” by Wu et al., has been improved. However, there are some concerns described below that should be addressed before publication.

#1: line 13-

It is described that “…each dimer has one carbohydrate recognition domain”. Since the NaGal-1 monomer has one carbohydrate domain, it should be changed, for example, to “…each ‘monomer’ has one carbohydrate recognition domain” or  “…each dimer has ‘two identical’ carbohydrate recognition domain”

#2: line 177- (Figure 7)

In Figure 7B, the concentration of competitive sugars is not clearly described. It should be described whether the higher concentration of competitive sugars is at the top of the figure or at the bottom.

#3: line 262-

As to galectin binding to LPS and PGN, it is described in the response letter “Sugar binding motifs were the key factor to perform function of galectin, H-NPR and W--E-R were directly involved in carbohydrate binding”. However, such an amino acid sequence motif is generally thought to be important for galectin binding to galactose or related saccharides (Check the first sentence of page 4 of the following manuscript or other appropriate paper: “Galectins as Molecular Targets for Therapeutic Intervention” https://pubmed.ncbi.nlm.nih.gov/29562695/). In addition, the structure and sugar component of LPS and PGN could be varied by the species, and their glycan structure could be important for the recognition by galectins. For example, it has been reported that some mammalian galectins bind to certain species of bacteria expressing human blood group antigen (“Innate immune lectins kill bacteria expressing blood group antigen” https://pubmed.ncbi.nlm.nih.gov/20154696/). Therefore, as to galectin binding to LPS and PGN, the sugar component and structure of them used in this study should be described and discussed at least.

Author Response

#1: line 13: It is described that “…each dimer has one carbohydrate recognition domain”. Since the NaGal-1 monomer has one carbohydrate domain, it should be changed, for example, to “…each ‘monomer’ has one carbohydrate recognition domain” or  “…each dimer has ‘two identical’ carbohydrate recognition domain”

Response: each subunit has one carbohydrate recognition domain.

#2: line 177- (Figure 7): In Figure 7B, the concentration of competitive sugars is not clearly described. It should be described whether the higher concentration of competitive sugars is at the top of the figure or at the bottom.

Response: We thank the reviewer’s question. From top to bottom are corresponding to high-to-low concentrations of competitive sugars: the initial concentration of PGN and LPS are 12.22 ng/mL, the initial concentration of D-galactose and lactose are 39.06 µg/mL, and then two-fold serial dilutions are made.

#3: line 262: As to galectin binding to LPS and PGN, it is described in the response letter “Sugar binding motifs were the key factor to perform function of galectin, H-NPR and W--E-R were directly involved in carbohydrate binding”. However, such an amino acid sequence motif is generally thought to be important for galectin binding to galactose or related saccharides (Check the first sentence of page 4 of the following manuscript or other appropriate paper: “Galectins as Molecular Targets for Therapeutic Intervention” https://pubmed.ncbi.nlm.nih.gov/29562695/). In addition, the structure and sugar component of LPS and PGN could be varied by the species, and their glycan structure could be important for the recognition by galectins. For example, it has been reported that some mammalian galectins bind to certain species of bacteria expressing human blood group antigen (“Innate immune lectins kill bacteria expressing blood group antigen” https://pubmed.ncbi.nlm.nih.gov/20154696/). Therefore, as to galectin binding to LPS and PGN, the sugar component and structure of them used in this study should be described and discussed at least.

Response: We thank the reviewer’s helpful suggestion. As suggested, we have made corresponding modification in the discussion and references.